# Numerical Evaluation Using the Finite Element Method on Frontal Craniocervical Impact Directed at Intervertebral Disc Wear

Alfonso Trejo-Enriquez *, Guillermo Urriolagoitia-Sosa *, Beatriz Romero-Ángeles * , Miguel Ángel García-Laguna, Martín Guzmán-Baeza, Jacobo Martínez-Reyes, Yonatan Yael Rojas-Castrejon, Francisco Javier Gallegos-Funes , Julián Patiño-Ortiz and Guillermo Manuel Urriolagoitia-Calderón

Instituto Politécnico Nacional, Escuela Superior de Ingeniería Mecánica y Eléctrica, Sección de Estudios de Posgrado e Investigación, Unidad Profesional Adolfo López Mateos Zacatenco, Lindavista, Ciudad de México C.P. 07320, Mexico; mgarcial2100@alumno.ipn.mx (M.Á.G.-L.); maguzmanb@ipn.mx (M.G.-B.); jmartinezr0617@ipn.mx (J.M.-R.); yrojasc1300@alumno.ipn.mx (Y.Y.R.-C.); fgallegosf@ipn.mx (F.J.G.-F.); jpatinoo@ipn.mx (J.P.-O.); gurriolagoitiac@ipn.mx (G.M.U.-C.)
* Correspondence: atrejoe2100tmp@alumnoguinda.mx (A.T.-E.); guiurri@hotmail.com (G.U.-S.); romerobeatriz97@hotmail.com (B.R.-Á.)

**Abstract:** Traumatic cervical pathology is an injury that emerges due to trauma or being subjected to constant impact loading, affecting the ligaments, muscles, bones, and spinal cord. In contact sports (the practice of American football, karate, boxing, and motor sports, among others), the reporting of this type of injury is very common. Therefore, it is imperative to have preventive measures so players do not suffer from such injuries, since bad practices or accidents can put their lives at risk. This research evaluated cervical and skull biomechanical responses during a frontal impact, taking into consideration injury caused by wear on the intervertebral disc. Intervertebral disc wear is a degenerative condition that affects human mobility; it is common in people who practice contact sports and it can influence the response of the cervical system to an impact load. The main objective of this work is to evaluate the effects caused by impact loading and strains generated throughout the bone structure (composed of the skull and the cervical spine). The numerical evaluation was developed using the finite element method and the construction of the biomodel from computational axial tomography. In addition, the numerical simulation allowed us to observe how the intervertebral disc's wear affected the cervical region's biomechanical response. In addition, a comparison could be made between a healthy system and a disc that had suffered wear. Finally, the analysis provided information valuable to understanding how an impact, force-related injury can be affected and enabled us to propose better physiotherapeutic procedures.

**Keywords:** numerical simulation; finite element method; biomodel; impact load

## 1. Introduction

Spinal cord injury is a problem that occurs in 80% of the entire general population (regardless of the job they perform). Principal activities causing these injuries are sports practice, working activities, and routine daily events [1,2]. However, people who practice a contact sport are at higher risk of developing cervical pathology. Among sports that stand out for producing this kind of injury is American football (where there is a 56% chance of developing a degenerative pathology) (Figure 1) [3]. Of all the injuries that can be caused, the one that stands out the most is cervical disc herniation in the lower area (C3–C7), which produces a fracture in the odontoid process [4,5]. The main reason for this is that the players constantly collide while practicing this sport. In addition, gym preparation for this sport includes weightlifting, which can overload participants' discs; with time, these discs can wear out, resulting in a disc rupture [6]. Disc rupture can be affected

by spine anomalies, for example, the traumatic ones caused by head collisions [7]. In addition, degenerative injuries occur due to the misuse of anti-inflammatory drugs, being overweight, and weight loss [8]. To be able to observe critical effects generated by cervical pathology, a craniocervical biomodel (consisting of the skull, cervical vertebrae C1–C5, intervertebral discs, and spinal cord) was developed, and finite-element-method numerical analysis was performed to ensure it represented the reality of the injury effects as closely as possible [9]. These injuries, in some cases, generated a disability. Pathologies in the cervical spine are expected since players are exposed to constant physical contact (the most common impacts are frontal ones) [10]. The injury mechanisms generated by impacts cause traumatism to the skull, and the cervical spine is exposed to hyperextension movements with lateral flexion, which causes wear to develop due to the force of the hoof impact [11]. The craniocervical frontal effect depends on the impact force, which can result in serious neck injuries. The severity of these injuries depends on age and the time of exposure to this activity [12], which is a measurement factor due to the probability of wear on the intervertebral discs. This can occur more in a player at a professional level than a player at a nonprofessional level [13].

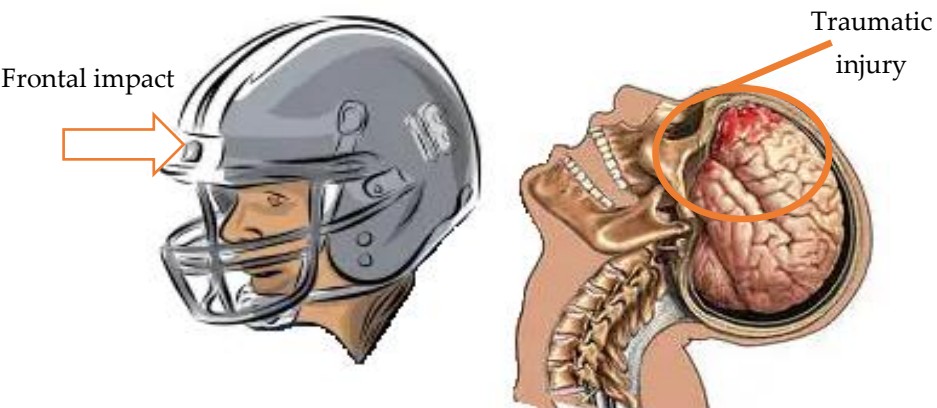

**Figure 1.** Frontal impact and possible effects.

Numerical analysis is a powerful tool for understanding the effects of frontal impact related to the biomechanical behavior of the neck and spine [14]. It is applied to simulate and predict the physical responses of these biological tissues under different loading conditions [15]. The implementation of this technology assists in evaluating the biomechanical behavior of biological systems through the use of computerized axial tomography. By performing a tomographical study, the biological tissue structure can be recreated in a 3D manner. This paper develops a complex biomodel by introducing the skull, cervical spine, intervertebral discs, and spinal cord [16]. Two cases of study are presented. The first study case considers a subject in a healthy condition, while the second study numerically simulates wear in the intervertebral disc [17,18]. This numerical analysis aimed to evaluate how intervertebral disc wear affects the biomechanical response of the neck and spine during a frontal impact. The described biomodels were analyzed by implementing the finite element method, which was used to simulate the behavior of tissues and bone structures [19]. The numerical analysis aimed to provide relevant data for the design of safety measures for players and for injury prevention in individuals with intervertebral disc wear. By better understanding the injury mechanisms and how they interact with disc wear, the development of effective strategies could be improved to minimize risks and progress safety for players playing this sport at a professional level [20]. Biomodels allow us to observe the behaviors of bone structures exposed to impact in a real situation since they are based on the use of tomography in medical diagnosis. However, biomodelling brings the scenario closer to reality by being three-dimensional because you can see the severity of the injury, which helps in proposing prevention or recovery treatment that can be performed in a personalized way. With the craniocervical biomodelling presented



in this paper, you can see the behavior of a joint complex before an impact where one can see the damage's severity before a surgical operation, thereby allowing one to work with rehabilitation and physiotherapeutic treatment. Analyzing the entire joint complex is crucial since it allows us to visualize its behavior in the event of an impact. Other authors have only analyzed the skull or the cervical spine, distancing their studies from accuracy. This is why, with cervical degenerative pathology, the study using the biomodel allows us to analyze existing prostheses with different biocompatible materials in order to optimize them in the future [21,22].

## 2. Methods

To conduct numerical analysis of frontal impact on the skull, a craniocervical biomodel was developed. For this study, a male patient, an American football player who was 1.85 m tall and weighed 120 kg, was selected. The patient underwent computational axial tomography to produce images of the human skull, cervical region, discs, and spinal cord. The tomography was imported in DICOM format to the SCAN IP computer program and displayed in grayscale in order to visualize the cortical and trabecular bone of each part of the bone system in the different sections, which were displayed in three windows (representing views in the coronal, axial, and sagittal axes) [23,24]. These views allowed us to delimit the area of interest and develop the bones of the biomodel (Figure 2).

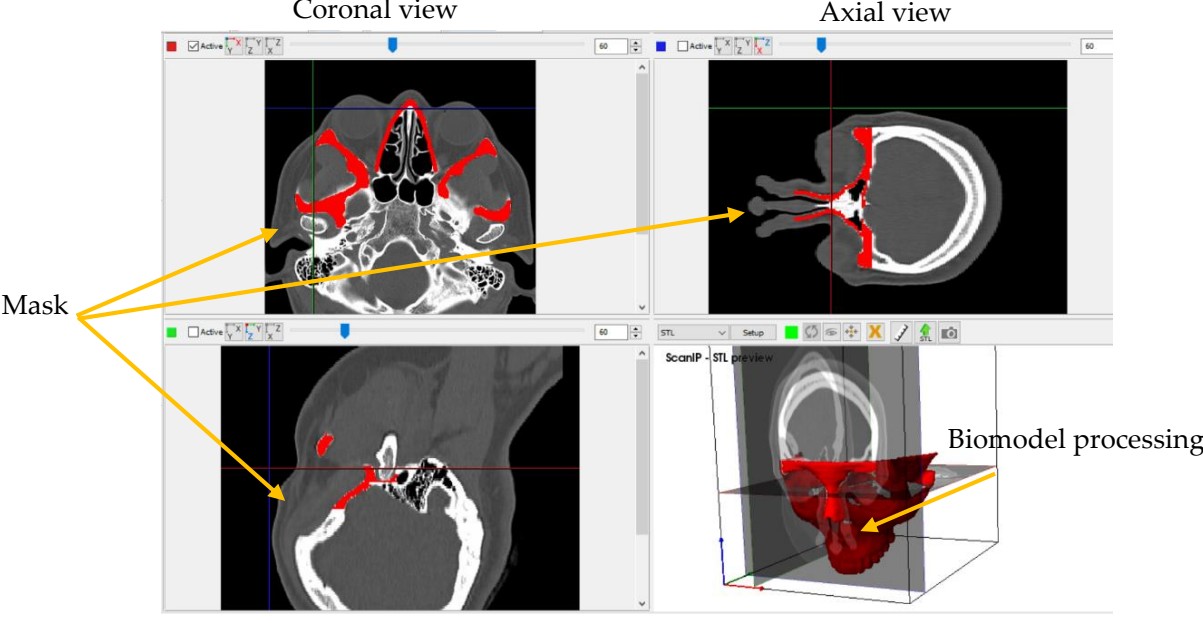

**Figure 2.** Patient's craniocervical computational tomography.

In Figure 3, the red mask represents the trabecular bone of the skull and the cervicals, yellow represents the cortical bone of the skull, blue represents the intervertebral discs, and pink represents the spinal cord. For the cervicals, C1 is represented by the green color, C2 the purple color, C3 the orange color, C4 the white color, and C5 the brown color. Once the masks were generated, they were exported in STL extension format and smoothing was applied to the surface of each of the components of the craniocervical system that made up the biomodel, which was developed through 3-Matic Medical software (a tool that allows the mesh to be corrected and simulates the wear on the intervertebral discs through material remotion) (Figure 4) [25].

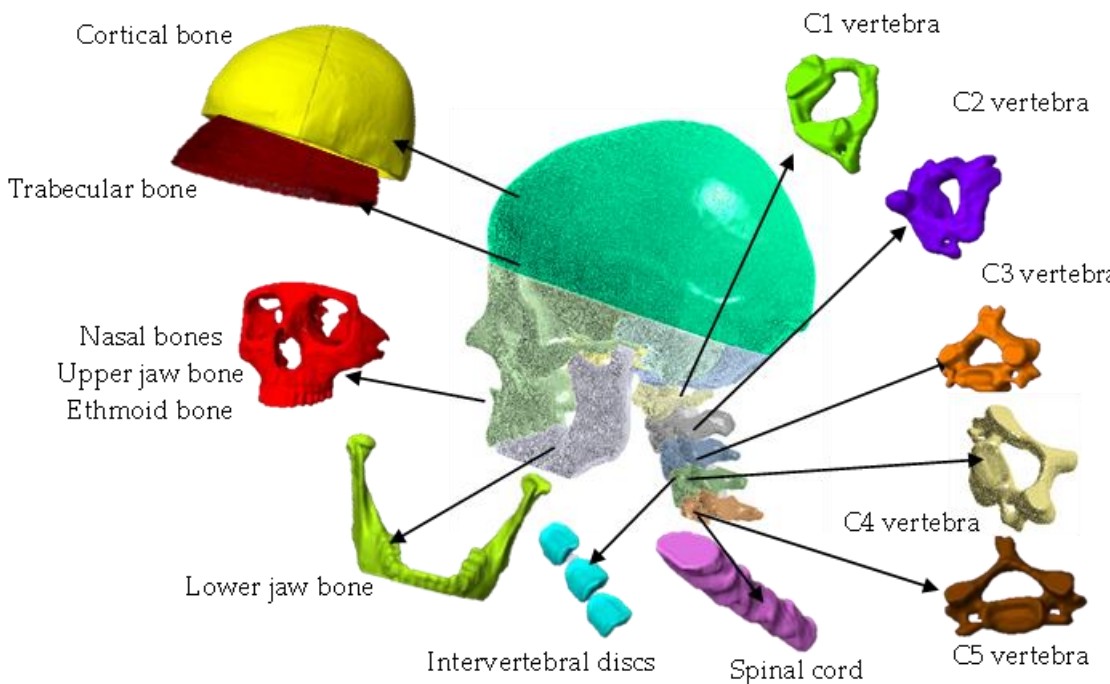

**Figure 3.** Biological system generated with masks.

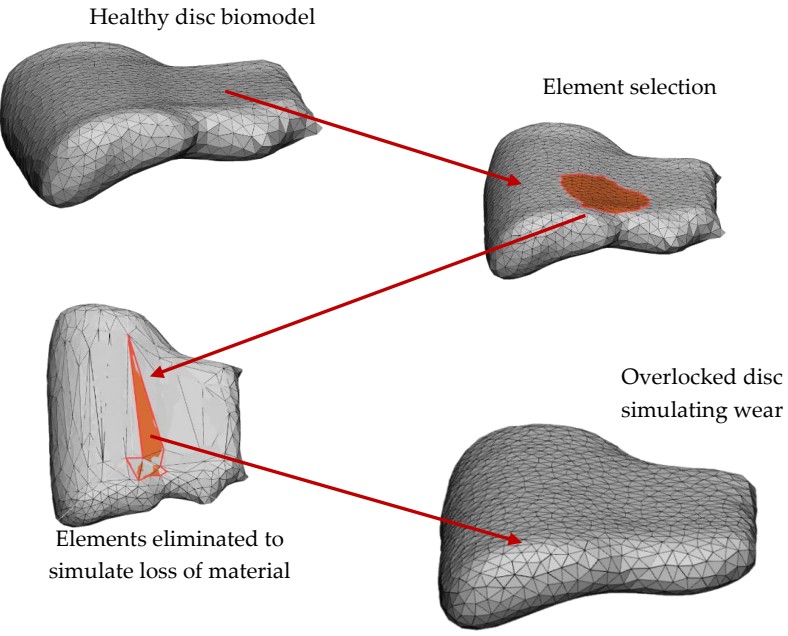

**Figure 4.** Wear process simulation on the intervertebral discs.

At the end of this process, biomodels of the cortical and trabecular bones (skull and cervical) were obtained along with ones for the bone marrow and intervertebral discs. Where a biomodel was considered to have high biofidelity and met the desired morphological characteristics, as the base with which it was performed was a tomography representing the craniocervical part of the bone system, the external dimensions of the contours of the bones (cortical and trabecular) were taken as references. Again, they were exported as STL files on which numerical evaluation was performed using the finite element method in the Ansys Workbench software [26].

*Materials*

Numerical analysis was performed by applying biomodeling, which was produced by computational tomography, and it was essential to assess the properties of the materials that describe the mechanical behavior of the biological tissues that were analyzed, which were as follows:

- Skull.
- Spinal cord.
- Cervical region: C1, C2, C3, C4, C5.
- Intervertebral disc.

The experimental analyses were the fundamental basis that supported the results obtained from the numerical studies. However, these results could be verified with significant technological development where economics are involved, saving time and material resources. For example, to understand the mechanical properties of the cervical and intervertebral disc, a numerical analysis of axial compression was carried out to support previous experimental research where the result was an approximation of 9.4%, which was acceptable [16]. Based on these results, the values considered for the cervical discs are presented in Tables 1 and 2 for the intervertebral discs. Table 3 shows the mechanical properties of the skull, which were obtained by numerical compression analysis [27].

**Table 1.** Mechanical properties assigned to the cervical bone [28].

| Properties | Cortical Bone | Trabecular Bone |
|---|---|---|
| Young´s modulus | 12,000 MPa | 100 MPa |
| Density | 1700 kg/m$^3$ | 0.14 g/cm$^3$ |
| Poisson ratio | 0.35 | 0.20 |

**Table 2.** Mechanical properties assigned to the intervertebral disc [29].

| Properties | Nucleus Pulposus | Annulus Fibrosus |
|---|---|---|
| Young´s modulus | 1 MPa | 8.4 MPa |
| Density | 997 kg/m$^3$ | 433 kg/m$^3$ |
| Poisson ratio | 0.40 | 0.35 |

**Table 3.** Mechanical properties are assigned to the skull bone [30].

| Properties | Cortical Bone | Trabecular Bone |
|---|---|---|
| Young´s modulus | 15,000 MPa | 200 MPa |
| Density | 1900 kg/m$^3$ | 430 g/cm$^3$ |
| Poisson ratio | 0.30 | 0.45 |

## 3. Numerical Analysis

For this work, two study cases were considered: a healthy case and a case where wear affected the intervertebral disc. The complex biomodel (the healthy and worn biomodel, respectively) was imported into the Ansys Workbench software, and the numerical simulation was carried out. The numerical analysis was a dynamic evaluation since the cervical spine was in motion, and the applied load was performed at high speed. Another aspect to consider was the properties of the previously explained materials, which based their studies on data in order to obtain the mechanical properties of each element that comprised the biological structure (the mechanical properties were declared). The biomodel considered 12 structural elements (bones and soft tissue) (Figure 3). Discretization was carried out in a semicontrolled manner by applying elements (producing 315,1321 nodes and 1,843,736 elements) (Figure 5). The material corresponding to each component of the biomodel was assigned.

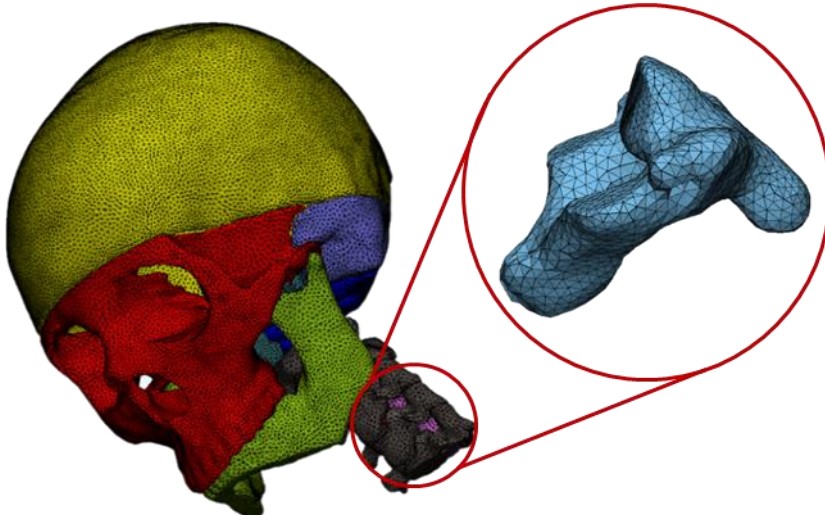

**Figure 5.** Discretized biomodel.

Then, the external agent conditions for the numerical simulation were introduced. First, the external agent applied for both study cases was considered as a pressure since the impact started in a specific area and the impact energy was distributed in a zone (Figure 6). Secondly, the boundary conditions (displacement and rotation restrictions) (Ux = Uy = Uz = 0, Rot XY = Rot YZ = Rot XZ = 0) were implemented in the lower zone of the C5 cervical region and spinal cord (Figure 7).

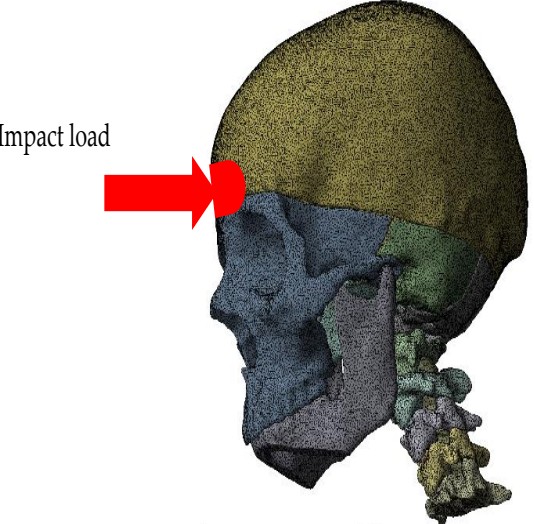

**Figure 6.** Loading application.

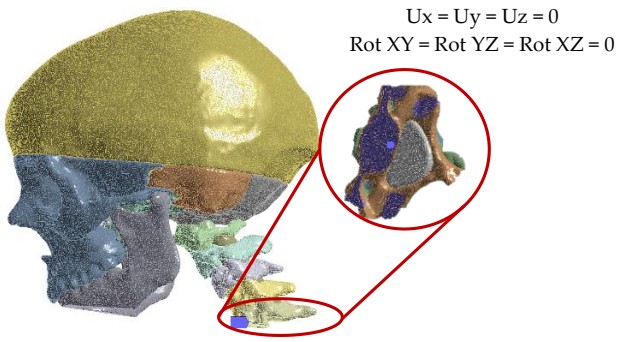

**Figure 7.** Application of the boundary conditions.

The National Standards Operations Committee for Athletic Equipment (NOCSAE) is a committee in the United States that certifies that the equipment used by players is the safest and takes care of the integrity of the person [30]. Every year, companies manufacturing American football helmets carry out evaluations with accelerometers inside the helmets to determine the impact force produced in a collision (around 30 g) [29]. Based on these results, for the numerical analysis, a force of 22 g was considered, which is the force exerted on a player in training camp, considering that they have experience collisions in practice than in a game. The conversion from g to m/s$^2$ is performed as follows:

$$22\,\text{g}\frac{9.81\,^\text{m}/\text{s}^2}{1\,\text{g}} = 215.82\,^\text{m}/\text{s}^2 \tag{1}$$

Knowing the acceleration and player's weight (120 kg), we calculate the impact load [31,32].

$$F = m\,a = (120\,\text{kg})\,(215.82\,\text{m/s}^2) = 25{,}898.4\,\text{N.} \tag{2}$$

The impact begins punctually, and the way it develops covers a specific area, which is 10 cm in diameter at the front of the skull (Figure 8), onto which pressure is applied to observe the energy of the impact dispersed throughout the skeletal system. A circle was chosen in order to see the behavior of the entire joint complex. If a sphere was considered, it had a penetration effect in the contact area where the forces were distributed in different ways. The pressure calculation was performed with the data obtained from the force and the implemented area [31].

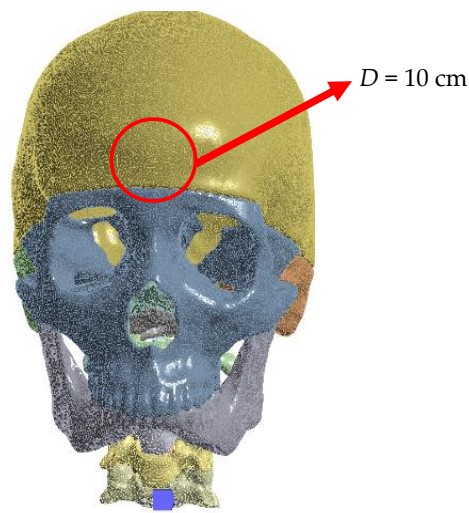

**Figure 8.** Impact area.

## 4. Results

It has been documented that American football players with disc wear develop severe headaches and suffer from reduced mobility [5, 6 y 17]. The main reason for this is that the disc no longer cushions the impact or helps movement. In addition, the disc, as it wears out, begins to move into the area of the spinal cord, exercising pressure on the nerve areas and the vertebrae, producing friction. This research work is based on two numerical analyses of a frontal impact. The first analysis considers the healthy intervertebral disc, and the second analyzes the intervertebral disc with wear. The most significant structural results in stress and displacement for both study cases are presented as follows (Figures 9–14) (Appendix A, Table A1).

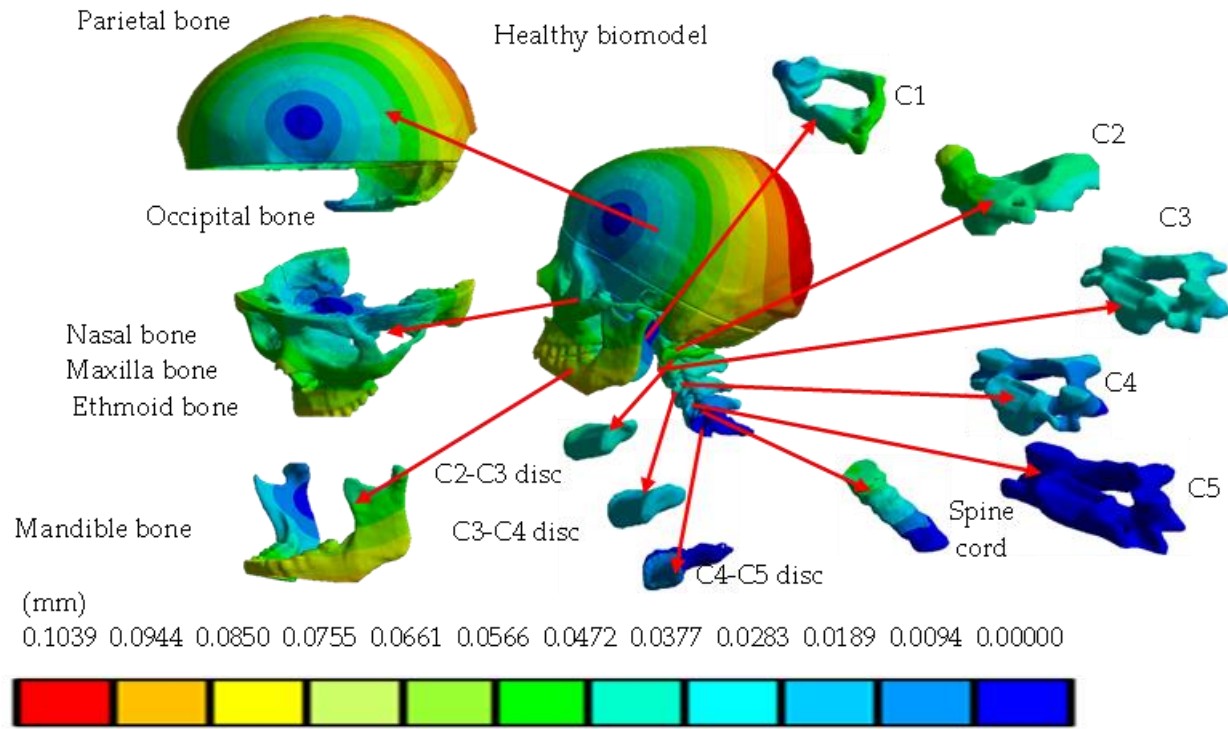

(mm)

0.1039 0.0944 0.0850 0.0755 0.0661 0.0566 0.0472 0.0377 0.0283 0.0189 0.0094 0.00000

**Figure 9.** Total displacement for healthy biomodel by components.

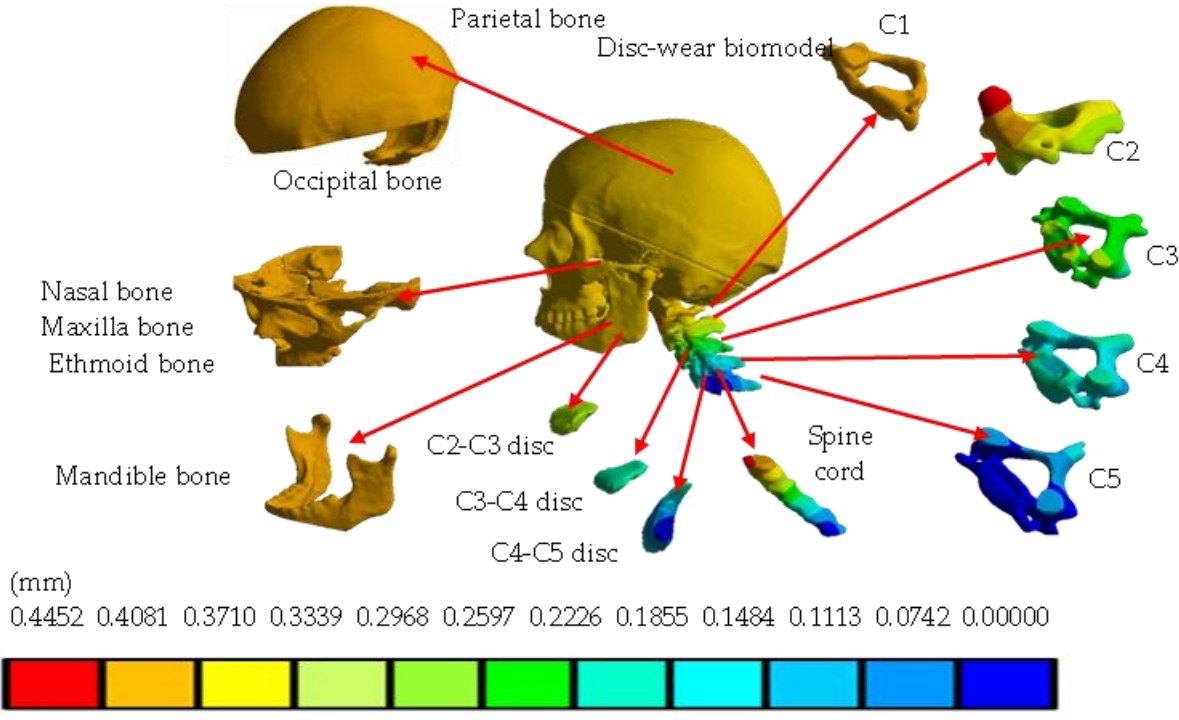

(mm)

0.4452 0.4081 0.3710 0.3339 0.2968 0.2597 0.2226 0.1855 0.1484 0.1113 0.0742 0.00000

**Figure 10.** Total displacement for disc-wear biomodel by components.

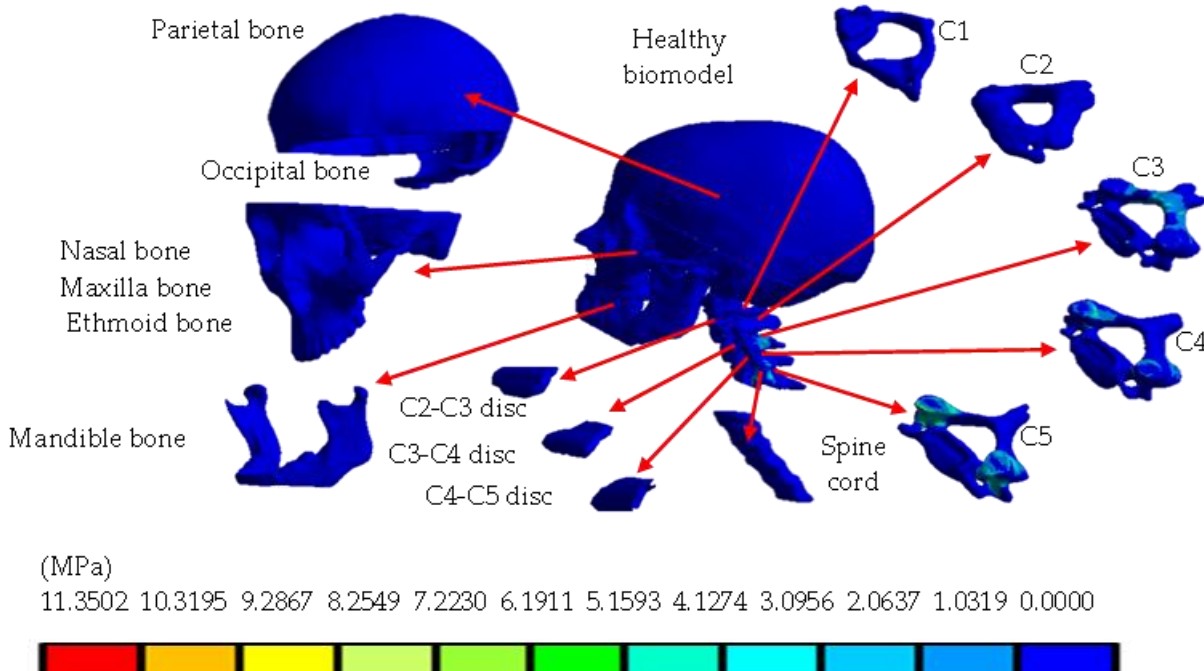

**Figure 11.** Von Mises stress for healthy biomodel by components.

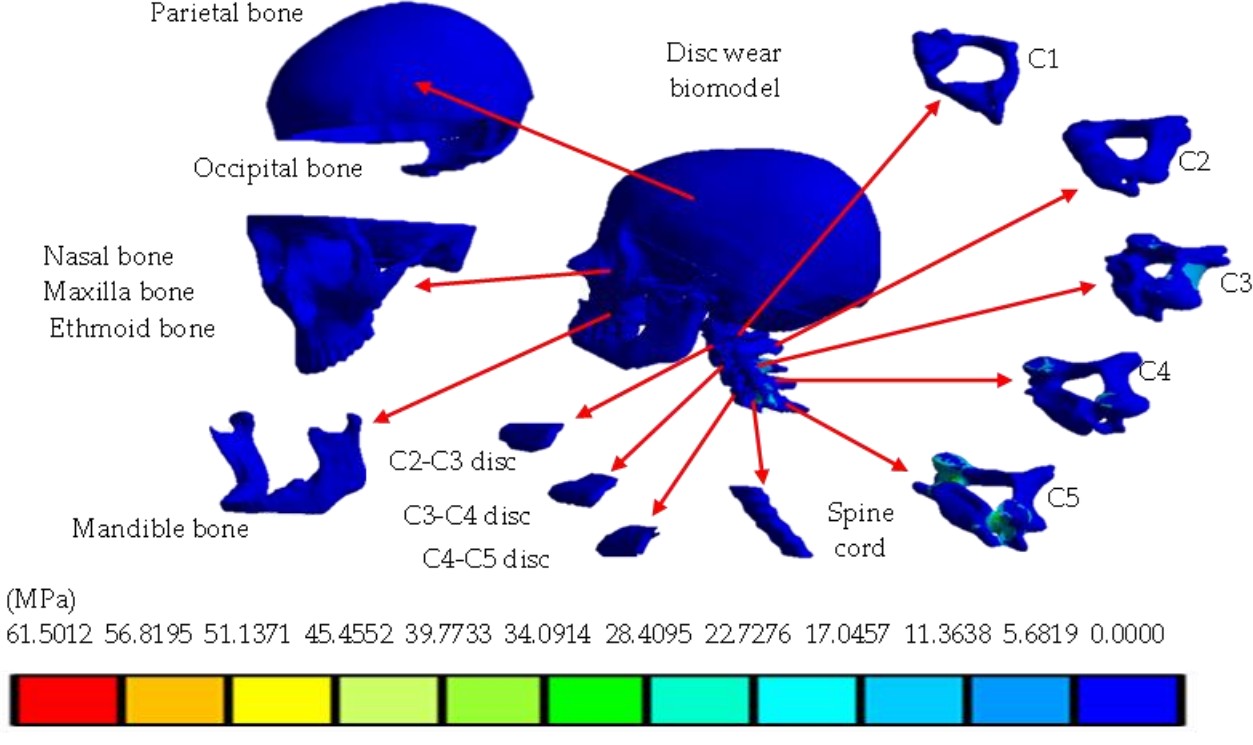

**Figure 12.** Von Mises stress for disc-wear biomodel by components.

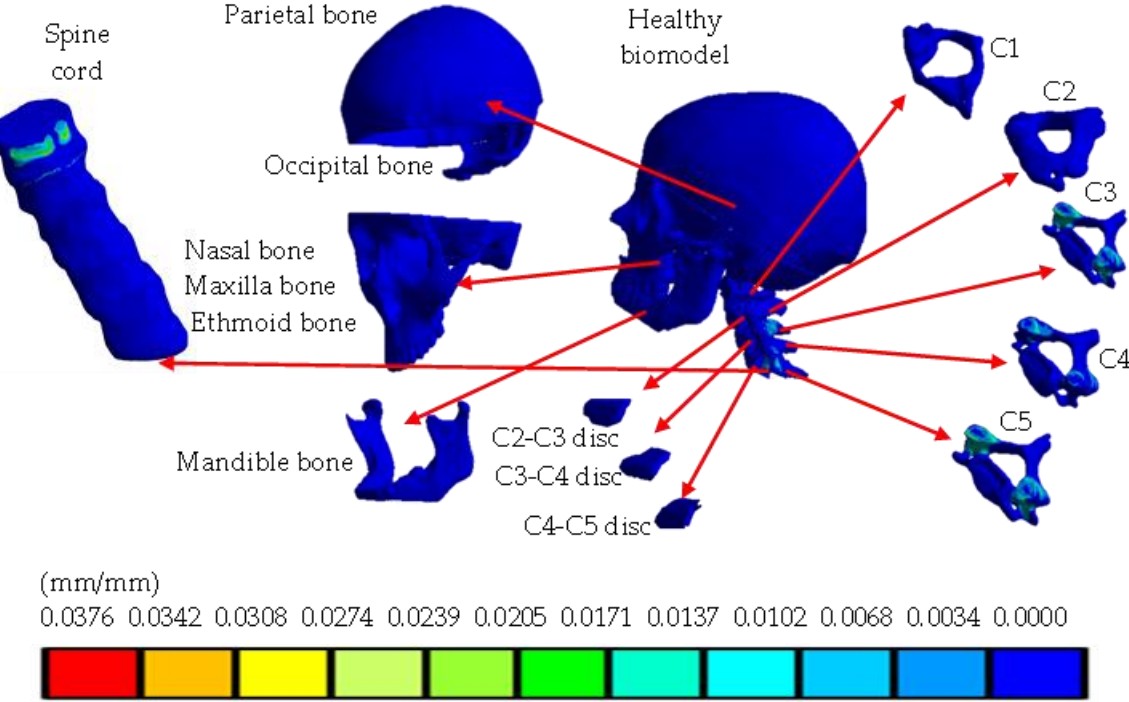

**Figure 13.** General equivalent strain for healthy biomodel by components.

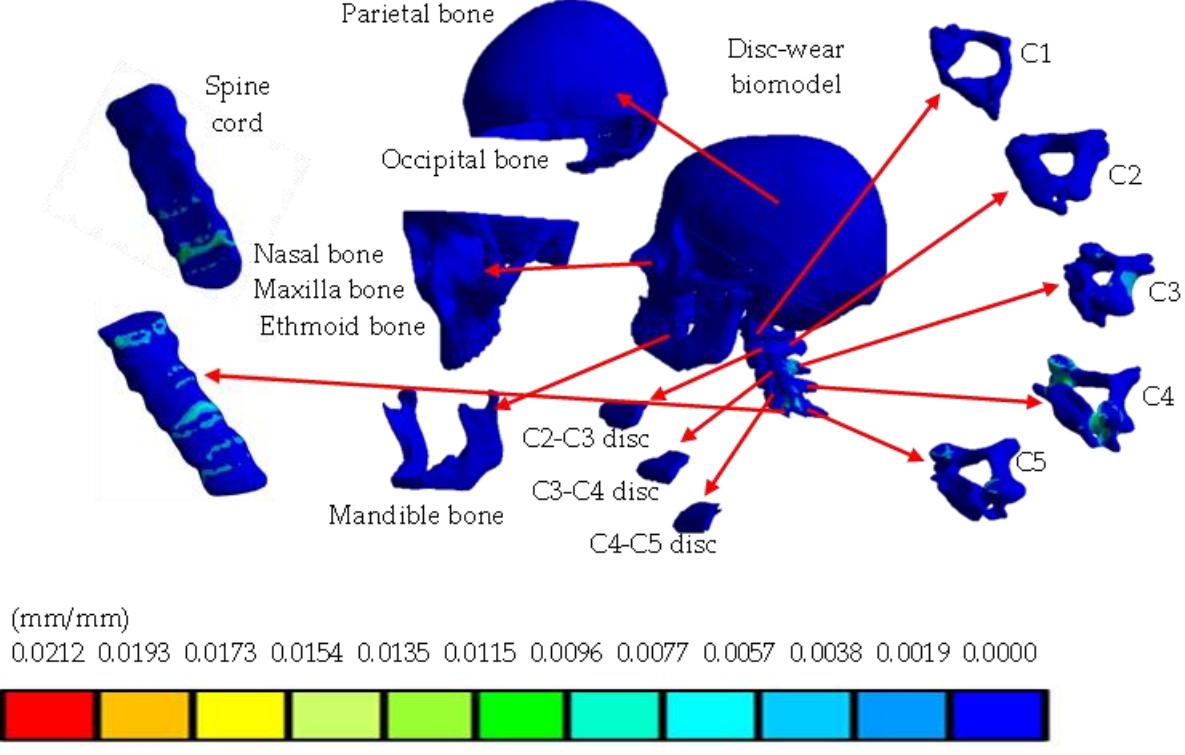

**Figure 14.** General equivalent strain for disc-wear biomodel by components.

Additionally, numerical analysis permitted the observation of substantial effects and estimated the damage produced by the disc's wear (Figures 15 and 16). In addition, a comparison between numerical cases being evaluated could be made. The numerical analyses were based on a free-body diagram to consider in a structural, mechanical manner the effects of the application of external agents and take into account the loading angles due to the displacement of the skull together with the cervical bones (Figures 17 and 18).

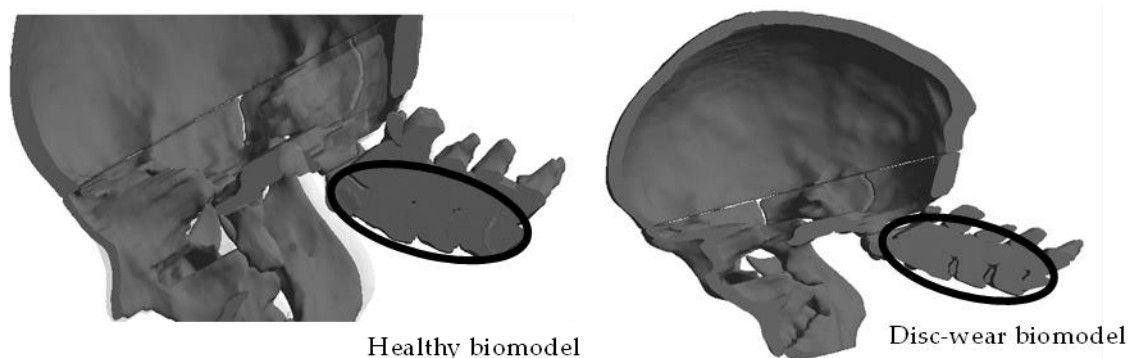

**Figure 15.** Comparison between cervical spine with healthy disc and cervical spine with disc wear.

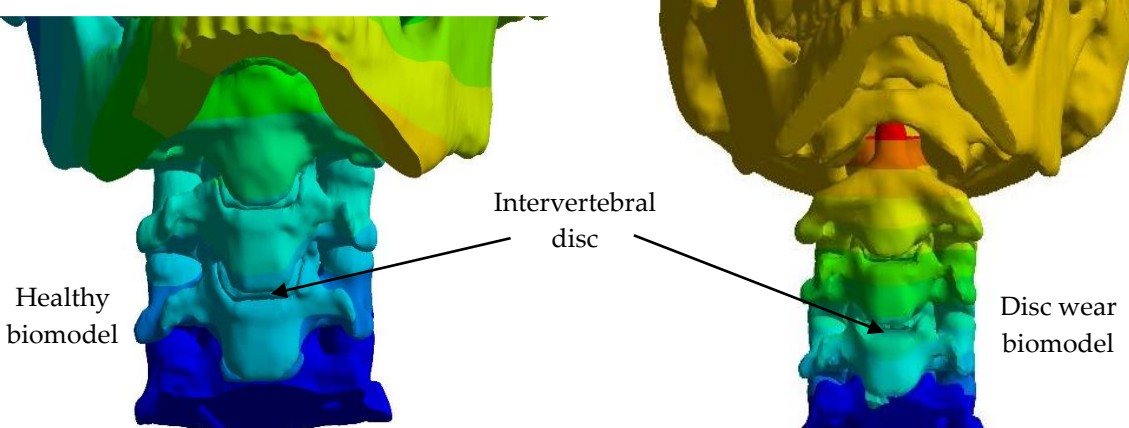

**Figure 16.** Zones implemented to visualize disc-wear nearness produced by impact loading.

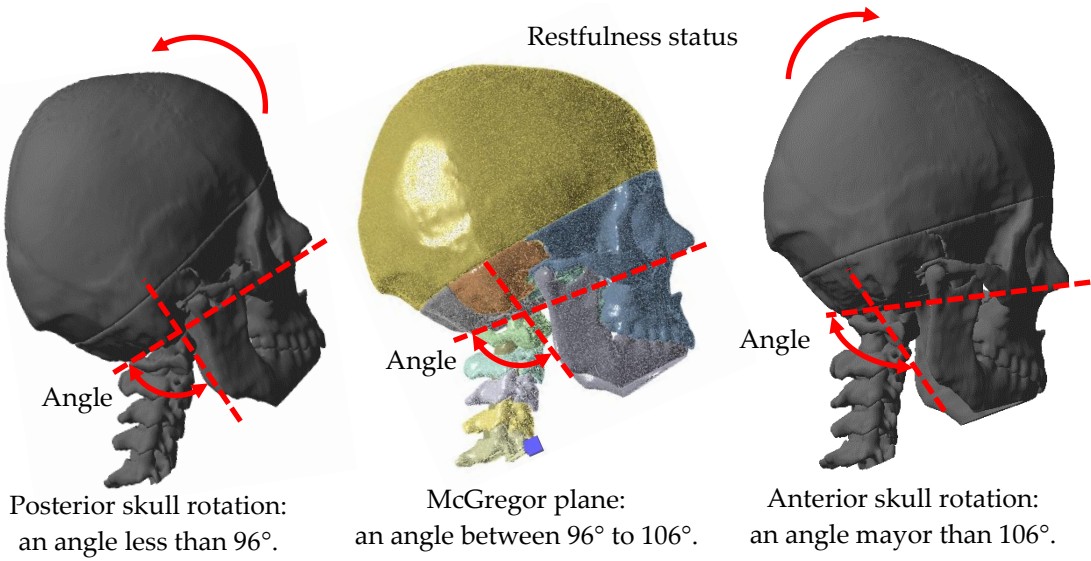

**Figure 17.** Comparison between free-body diagrams related to skull rotation angles that produce commotion.

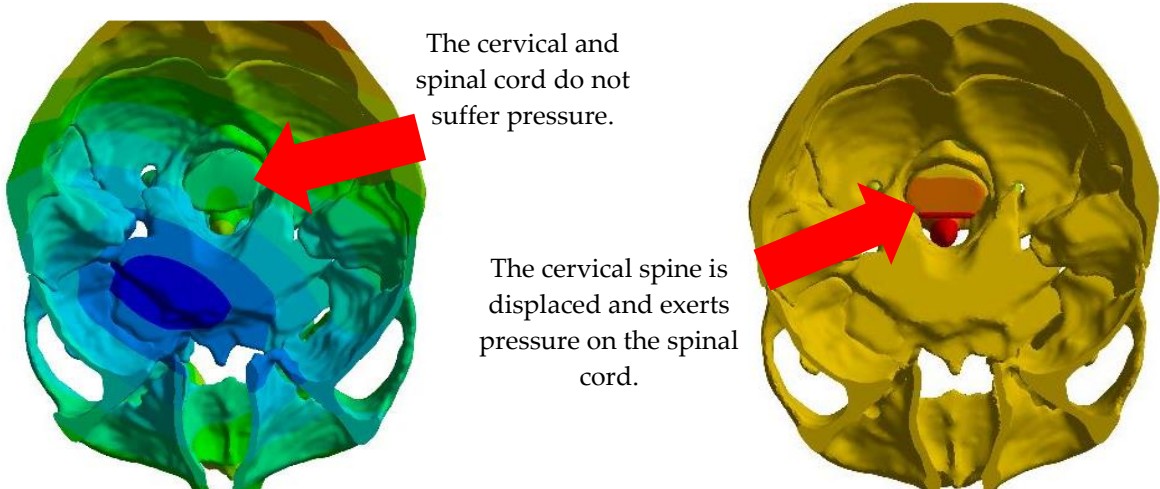

**Figure 18.** Graphical representation of discomfort effect when there is intervertebral disc wear.

## 5. Discussion

Even though computational technological development has opened the door for the development and representation of biological tissues in a three-dimensional way, it has not yet been possible to design a 100% faithful representation that meets the real morphological characteristics of a bone structure. The methodology applied for the development of the biomodel presented in this research work is considered to be of high biofidelity since the design is based on tomography to represent the biological tissue of the area of interest, which adapts to the contour of the structure, producing a biomodel that complies with the real morphology of the patient without compromising his physical integrity. This type of methodology is an auxiliary tool for designing complex biomodels with different biological tissues acting together. Having a more realistic image helps with the complexity of the implemented craniocervical biomodel since the research contemplates the development of a criterion that demonstrates, according to the displacements, how the impact energy is distributed throughout the entire joint complex. Comparisons can be made between the following scenarios: when the intervertebral disc has healthy conditions and when there is degenerative disc pathology. In addition, observations using unit deformations and stresses in the area most susceptible to injury can be performed. The remarkable thing about this research is that the results obtained could be beneficial for developing or improving preventive treatments or surgical procedures from a medical point of view. In addition, with this type of numerical analysis, intervertebral disc prostheses could be developed and optimized, considering different biocompatible materials and making comparisons in order to select the one that best suits the patient's conditions. It is essential to consider the entire joint complex because the impact begins at a point with a specific area and, from there, that impact energy is transmitted so that the work is carried out together, for this case study, in the skull. Hence, the cervical discs cushion the impact and one can observe their behavior with the spinal cord and, when you have the degenerative disc pathology, where that herniation occurs. On the other hand, other authors have only analyzed the skull, where the impact effect cannot be appreciated because the cervical and intervertebral discs with the spinal cord are missing. Finally, other authors have also only analyzed the cervical spine without all the elements that comprise it, and the skull, to show this effect [21,22].

## 6. Conclusions

With the results obtained in the numerical study, you can see the affected areas; you can deduce the reasons why players have symptoms of back and neck pain and, in very critical cases, severe headaches that limit their activities. Since the maximum efforts occur in susceptible areas of the vertebrae, more specifically in C3 to C5, where an isochromatic change can be seen in each structure, in addition to the displacement, it is observed how

the part of the bone marrow is compressed, which causes the symptoms that directly affect the head, such as headaches. For analysis of the wear of the disc, it is observed how a space is generated that makes the cervical ones closer together, causing friction between both biological structures. With the effect of the external agent, the worn disc causes the cervical region to compact along its longitudinal axis without allowing movement or cushioning, so the most significant energy is transmitted directly to the skull as the process generates pressure on C1, limiting the patient's activities. It is essential to verify how the affected areas change with disc wear to better understand what causes damage to the bone structure. Research could also focus on supporting the search for feasible solutions to treat diseases or injuries caused by this type of physical activity, emphasizing this discipline and all those that involve high-impact physical contact. Even simple activities performed in daily activities, such as driving, are at risk of causing frontal impact when a road accident occurs. Obtaining a biomodel that represents the components of the skull and neck (which make up the craniocervical structure) that contains a high biofidelity of each element of the system will produce an analysis close to reality, which allows the evaluation of possible symptoms presented by a healthy biological group when having intervertebral disc wear, thereby assisting the health sector in providing a better structural mechanobiological understanding from the point of view of classical mechanics. This promotes the design of new surgical treatments and allows scholars to propose new rehabilitation methods for patients with this pathology.

**Author Contributions:** Conceptualization, A.T.-E., G.U.-S., and B.R.-Á.; methodology, A.T.-E., G.U.-S., B.R.-Á., and G.M.U.-C.; validation, G.U.-S., A.T.-E., B.R.-Á., M.Á.G.-L., and M.G.-B.; formal analysis, A.T.-E., G.U.-S., B.R.-Á., and F.J.G.-F.; investigation, A.T.-E., G.U.-S., B.R.-Á., and G.M.U.-C.; resources, A.T.-E., G.U.-S., and B.R.-Á.; writing—original draft preparation, A.T.-E., G.U.-S., and B.R.-Á.; writing—review and editing A.T.-E., G.U.-S., B.R.-Á., and J.M.-R.; visualization, A.T.-E., G.U.-S., B.R.-Á., and Y.Y.R.-C.; supervision, A.T.-E., G.U.-S., and B.R.-Á.; project administration, A.T.-E., G.U.-S., B.R.-Á., and J.P.-O. All authors have read and agreed to the published version of the manuscript.

**Funding:** This research received no external funding.

**Institutional Review Board Statement:** Not applicable.

**Informed Consent Statement:** Not applicable.

**Data Availability Statement:** Not applicable.

**Acknowledgments:** The authors gratefully acknowledge the Instituto Politécnico Nacional and the Consejo Nacional de Humanidades, Ciencias y Tecnologías, for the support of this research.

**Conflicts of Interest:** The authors declare no conflict of interest.

## Appendix A

**Table A1.** Summary of general results of numerical evaluation.

| Concept | Healthy Condition | | Disc-Wear Condition | |
|---|---|---|---|---|
| | Minimal | Maximum | Minimal | Maximum |
| Total displacement (mm) | 0 | 0.1039 | 0 | 0.4452 |
| Displacement X axis (mm) | −0.0444 | 0.0653 | −0.02043 | 0.00941 |
| Displacement Y axis (mm) | −0.0724 | 0.0882 | −0.3297 | 0.0911 |
| Displacement Z axis (mm) | −0.0727 | 0.0531 | −0.3163 | 0.1465 |
| Total strain (mm/mm) | 0 | 0.0376 | 0 | 0.0212 |
| Strain X axis (mm/mm) | −0.0174 | 0.0082 | −0.0042 | 0.0031 |
| Strain Y axis (mm/mm) | −0.0204 | 0.0133 | −0.0123 | 0.0044 |
| Strain Z axis (mm/mm) | −0.0215 | 0.0108 | −0.0117 | 0.0070 |
| Von Mises stress (MPa) | 0 | 11.3502 | 0 | 62.501 |

**Table A1.** *Cont.*

| Concept | Healthy Condition | | Disc-Wear Condition | |
|---|---|---|---|---|
| | Minimal | Maximum | Minimal | Maximum |
| Maximum principal stress (MPa) | −4.7549 | 12.6048 | −14.507 | 59.984 |
| Minimum principal stress (MPa) | −15.5285 | 3.4748 | −68.268 | 6.4836 |
| Maximum shear stress (MPa) | 0 | 5.8328 | 0 | 34.632 |
| Nominal stress X axis (MPa) | −8.0763 | 5.8252 | −24.959 | 22.713 |
| Nominal stress Y axis (MPa) | −11.9718 | 7.1022 | −31.995 | 24.682 |
| Nominal stress Z axis (MPa) | −14.9241 | 11.2437 | −65.264 | 55.777 |
| Shear stress XY plane (MPa) | −2.0796 | 2.3934 | −11.901 | 12.005 |
| Shear stress YZ plane (MPa) | −4.1420 | 5.1737 | −25.828 | 25.919 |
| Shear plane XZ plane (MPa) | −3.8517 | 3.3848 | −11.612 | 21.842 |

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
