# Peer review of "Numerical Evaluation Using the Finite Element Method on Frontal Craniocervical Impact Directed at Intervertebral Disc Wear"

_applsci, doi:10.3390/app132111989_

Round 1
Reviewer 1 Report
Comments and Suggestions for Authors
1. The materials parameters of the intervertebral disc should be updated, for example, the hyperelastic should be used for the Nucleus pulposus. I want to know the predicted results when the hyperelastic material parameter is used.
2. The accuracy of the FE model should be verified in more detail.
3. The ethics approval was not included about the volunteer.
4. The figures should be decreased, not more than 10.
5. The results and discussion should be rewriteen, because the simulation results and implications of this paper are not clearly explained.
6. The abstract and conslusion may be decreased appropriately.
Comments on the Quality of English LanguageThe English Language is well.
Author Response
|
Comments 1: The materials parameters of the intervertebral disc should be updated, for example, the hyperelastic should be used for the Nucleus pulposus. I want to know the predicted results when the hyperelastic material parameter is used.
|
|
Response 1: Thank you for pointing this out. I agree with this comment. Therefore, Because the system presents a problem in the continuity of discretization by combining different mechanical properties, the linear parameters for the intervertebral disc were preserved. Although you can choose to perform the analysis in a new computer program, this would take a little more time, which could affect the review process of this research work.
|
|
Comments 2: The accuracy of the FE model should be verified in more detail |
|
Response 2: Thank you for pointing this out. I agree with this comment. Therefore, we have Herself-specific updated text in the manuscript |
|
Comments 3: The ethics approval was not included about the volunteer. |
|
Response 3: Thank you for pointing this out. I agree with this comment. Therefore, we have Herself-specific updated text in the manuscript
|
|
Comments 4: The figures should be decreased, not more than 10. |
|
Response 4: Thank you for pointing this out. I agree with this comment. Therefore, we have Herself-specific updated text in the manuscript |
|
Comments 5: The results and discussion should be rewriteen, because the simulation results and implications of this paper are not clearly explained. |
|
Response 5: Thank you for pointing this out. I agree with this comment. Therefore, we have Herself-specific updated text in the manuscript |
|
Comments 6: The abstract and conclusion may be decreased appropriately. |
|
Response 6: Thank you for pointing this out. I agree with this comment. Therefore, we have Herself-specific updated text in the manuscript |

Reviewer 2 Report
Comments and Suggestions for Authors
1. The novel aspects of this work should be clearly presented in a separate paragraph at the end of the introduction section.
2. "Poisson ratio" > "Poisson's ratio."
3. The considered mechanical properties ignore the potential influence of sex and age on each individual, as these factors can lead to changes in all mechanical properties.
4. The reason behind choosing a circle with a 10cm diameter as the impact area should be explained. Alternatively, it is suggested that a numerical contact with a spherical impactor could provide a more realistic representation of the impact area.
5. Figures 12 and 13 should be revised by changing the scale bar to logarithmic mode, which will effectively demonstrate the high variability of induced stress.
6. To facilitate comparison between the finite element (FE) results and critical values, information about damage should be included in Tables 1 to 3. This will involve adding critical stresses to the table.
7. More comprehensive elaboration on the results is required. For instance:
- A graph should be drawn to illustrate the total deformation over time for both cases.
- A path should be used to depict the stress or deformation distribution in the critical area of the model.
Comments on the Quality of English Language-
Author Response
|
Comments 1: The novel aspects of this work should be clearly presented in a separate paragraph at the end of the introduction section.
|
|
Response 1: Thank you for pointing this out. I agree with this comment. Therefore, we have Herself-specific updated text in the manuscript.
|
|
Comments 2: Poisson ratio" > "Poisson's ratio." |
|
Response 2: Thank you for pointing this out. I agree with this comment. Therefore, we have Herself-specific updated text in the manuscript |
|
Comments 3: The considered mechanical properties ignore the potential influence of sex and age on each individual, as these factors can lead to changes in all mechanical properties. |
|
Response 3: Thank you for pointing this out. I agree with this comment. Therefore, we have The characteristics of the individual are omitted from the text so that it does not interfere with the mechanical properties designated for this case study since the purpose of the case study is to obtain indicators of the areas that are affected in the biomodel caused by a frontal impact. |
|
Comments 4: The reason behind choosing a circle with a 10cm diameter as the impact area should be explained. Alternatively, it is suggested that numerical contact with a spherical impactor could provide a more realistic representation of the impact area. |
|
Response 4: Thank you for pointing this out. I agree with this comment. Therefore, we have Herself-specific updated text in the manuscript. The selection is because the uniform impact effect in a given area is sought. On the other hand, if the analysis were carried out with a spherical impactor, there would be a penetration effect in the contact area where the forces are distributed in different ways, and the objective is to observe the behavior of the entire joint complex. |
|
Comments 5: Figures 12 and 13 should be revised by changing the scale bar to logarithmic mode, effectively demonstrating the high variability of induced stress. |
|
Response 5: Thank you for pointing this out. I agree with this comment. Therefore, we have Herself-specific updated text in the manuscript. Another reviewer observed that the images should be reduced, so it was decided to omit Figure 12 and keep Figure 13, representing the same thing. They were just different ways of looking at the results. |
|
Comments 6: To facilitate comparison between the finite element (FE) results and critical values, information about damage should be included in Tables 1 to 3. This will involve adding critical stresses to the table. |
|
Response 6: Thank you for pointing this out. I agree with this comment. Therefore, we have Herself-specific updated text in the manuscript |
|
Comments 7: More comprehensive elaboration on the results is required. For instance:
- A graph should be drawn to illustrate the total deformation over time for both cases.
- A path should be used to depict the stress or deformation distribution in the critical area of the model. |
|
Response 7: Thank you for pointing this out. I agree with this comment. Therefore, we have Because the analysis is linearly elastic, the relationship taken is linear. It depends on the stress against the unit deformation and the fact that the analysis is structurally static. |

Reviewer 3 Report
Comments and Suggestions for Authors
1. Please insert the references for the data in the table 1,2,3
2. Please compare the obtained data with others from the literature in the section "5. Discussion", for a better understanding and convincing data.
3. Reformulate the conclusions in general findings, not discussion the figures 9, 18, 21. Move all the remarks from the conclusions in the section 5. Discussion.

Author Response
|
Comments 1: Please insert the references for the data in the table 1,2,3.
|
|
Response 1: Thank you for pointing this out. I agree with this comment. Therefore, we have Herself-specific updated text in the manuscript.
|
|
Comments 2: Please compare the obtained data with others from the literature in the section "5. Discussion", for a better understanding and convincing data. |
|
Response 2: Thank you for pointing this out. I agree with this comment. Therefore, we have Herself-specific updated text in the manuscript |
|
Comments 3: Reformulate the conclusions in general findings, not discussion the figures 9, 18, 21. Move all the remarks from the conclusions in the section 5. Discussion. |
|
Response 3: Thank you for pointing this out. I agree with this comment. Therefore, we have Herself-specific updated text in the manuscript |

Round 2
Reviewer 1 Report
Comments and Suggestions for Authors
It can be accepted.
Comments on the Quality of English LanguageThe Quality of English Language is well.
Author Response
Thank you for pointing this out. I agree with this comment. Therefore, we have
Herself-specific updated text in the manuscript